



# Background conditions for an urban greenhouse gas network in the Washington, D.C. and Baltimore metropolitan region

Anna Karion[1], Israel Lopez-Coto[2], Sharon M. Gourdji[1], Kimberly Mueller[1], Subhomoy Ghosh[1,3], William Callahan[4], Michael Stock[4], Elizabeth DiGangi[4], Steve Prinzivalli[4], James Whetstone[1]

[1]Special Programs Office, National Institute of Standards and Technology, Gaithersburg, MD, 20899, USA
[2]Fire Research Division, National Institute of Standards and Technology, Gaithersburg, MD, 20899, USA
[3]Center for Research Computing, University of Notre Dame, South Bend, IN, 46556, USA
[4]Earth Networks, Germantown, MD, 20876, USA

*Correspondence to*: Anna Karion (anna.karion@nist.gov)

**Abstract.** As city governments take steps towards establishing emissions reduction targets, the atmospheric research community is increasingly able to assist in tracking emissions reductions. Researchers have established systems for observing atmospheric greenhouse gases in urban areas with the aim of attributing greenhouse gas concentration enhancements (and thus, emissions) to the region in question. However, to attribute enhancements to a particular region, one must isolate the component

of the observed concentration attributable to fluxes inside the region by removing the background, which is the component due to fluxes outside. In this study, we demonstrate methods to construct several versions of a background for our carbon dioxide and methane observing network in the Washington, DC and Baltimore, MD metropolitan region. Some of these versions rely on transport and flux models, while others are based on observations upwind of the domain. First, we evaluate the backgrounds in a synthetic data framework, then we evaluate against real observations from our urban network. We find

that backgrounds based on upwind observations capture the variability better than model-based backgrounds, although care must be taken to avoid bias from biospheric carbon dioxide fluxes near background stations in summer. Model-based backgrounds also perform well when upwind fluxes can be modeled accurately. Our study evaluates different background methods and provides guidance determining background methodology that can impact the design of urban monitoring networks.

## 25  1 Introduction

In efforts to increase sustainability and address climate change, governments, private entities, and other stakeholders are tracking their greenhouse gas (GHG) emissions over time. Atmospheric observations have a crucial role to play in this effort, as they have the potential to provide a useful tool for assessing the effectiveness of emissions mitigation efforts. Urban atmospheric GHG monitoring networks have proliferated in the past decade, established by the carbon cycle research



community to assess the ability of such networks to detect trends and anomalies in urban emissions (Mitchell et al., 2018; Sargent et al., 2018; Lauvaux et al., 2020). Emissions estimates from such atmospheric observations rely on separating observed concentrations into two components: the concentration in the air entering the study domain and the enhancements in concentration attributable to emissions within the domain. This enhancement isolation is necessary for analysis whether it be for formal statistical inverse modeling of surface fluxes or for unbiased trend detection. In urban areas, background

determination is often difficult given the typically smaller study domain and the temporal and spatial variability of the background conditions relative to regional or global studies (Mueller et al., 2018; Balashov et al., 2020; Xueref-Remy et al., 2018).

Previous GHG studies in urban regions have utilized observations from a variety of platforms, including aircraft, ground-based column instruments, satellites, and in-situ stationary locations (such as rooftops or towers). Different approaches have been

used to isolate the background from observed concentrations from any of these platforms in order to perform analysis on enhancements. Urban analyses based on ground-based in-situ GHG observations often establish background concentrations using measurements from stations outside the urban domain, either upwind (often filtering data for a given wind sector) or in an area far from urban emissions. Sometimes these are from observations from a remote or baseline station, such as a mountain top or off-shore location (Mitchell et al., 2018; Verhulst et al., 2017). These background measurements are filtered for clean

conditions, to remove pollution events for example. A lowest percentile method has also been used as background, e.g., the lowest 5% of measurements during a certain time period, or a network-wide minimum value (Shusterman et al., 2016; Ammoura et al., 2016). In many studies, observations from a station that is upwind given daily meteorological conditions are used for background (Xueref-Remy et al., 2018; Lauvaux et al., 2016; Breon et al., 2015; Balashov et al., 2020), most often using observations from the same time of day as the urban station. A recent study of $CO_2$ in Boston used a more complex back-

trajectory-based method to sample the upwind station (Sargent et al., 2018). Just as for other methods, the background could also be optimized along with the urban fluxes within the inverse analysis (Nickless et al., 2018).

The goal of this study is to construct and evaluate a background for the Northeast Corridor Washington DC/Baltimore tower-based urban network described in Karion et al. (2020). We investigate many of the methods mentioned above, with some exceptions: we do not investigate the baseline/remote station, the low-percentile, or the optimized background methods. The

Washington/Baltimore region is downwind of many large flux regions (both anthropogenic and biospheric), and previous work has shown large synoptic variability in the background for the urban area (Mueller et al., 2018), so the use of a remote station or a lowest percentile method is not likely to produce an accurate representation of background variability. Optimizing the background in an inversion framework along with fluxes could be an option for our domain, but we do not perform an inverse analysis here. Instead, we present some background options that could be used as initial guesses, or priors, in a Bayesian

framework for optimization in the future.



Although the analysis we present is focused on carbon dioxide ($CO_2$) and methane ($CH_4$) in the Washington DC/Baltimore urban domain, we expect many of the overall methods for background estimation and evaluation explored in this study to be extensible to other urban or regional networks. In Sect. 2, we outline the methods for the study, including how we determine background values for the Washington DC/Baltimore network. In Sect. 3, we perform a synthetic data analysis to evaluate

$CO_2$ biases in three methods that use upwind observations from surface stations near the domain edge. We use the synthetic experiment to determine the best way to use these upwind observations. In Sect. 4 we evaluate $CO_2$ and $CH_4$ background time series constructed in different ways, including methods that rely on modeled upwind fluxes, against observations and compare their performance. Sect. 5 includes discussion of the results, and Sect. 6 makes conclusions and recommendations.

## 2 Methods

### 2.1 Definition of domain and background

Here we define the background for a given urban measurement to be the mole fraction that would be observed at that location and time in the absence of any GHG fluxes inside the domain of interest. Therefore, we separate the $CO_2$ or $CH_4$ mole fraction measured at each station ($y$) as a combination of a background ($y_{BG}$) and an enhancement from fluxes within the domain of interest ($y_{enh}$):

$$y = y_{BG} + y_{enh} \tag{1}$$

We note that $y_{enh}$ may be positive or negative, depending on the direction of fluxes in the domain. For our study, this domain is an area approximately 140 km by 135 km surrounding the cities of Washington, DC, and Baltimore, MD, and encompassing their larger metropolitan areas (Figure 1). A network of observation stations on existing towers has been established by Earth Networks and NIST comprising 11 urban towers, i.e., towers situated inside the domain, and 3 background towers, i.e., towers

situated near the edges (TMD, SFD, BUC in Figure 1). Locations were determined by network design studies (Lopez-Coto et al., 2017; Mueller et al., 2018). Details on the atmospheric $CO_2$ and $CH_4$ mole fraction measurements from this network are found in Karion et al. (2020). In this study we use observations from the six urban sites in Figure 1, as we focus on November 2016 through October 2017, when only these six had been established. In this work, $CO_2$ measurements are given as dry air mole fractions, with units of mmol mol$^{-1}$, or parts per million (ppm); $CH_4$ dry air mole fractions are in units of nmol mol$^{-1}$, or

parts per billion (ppb).



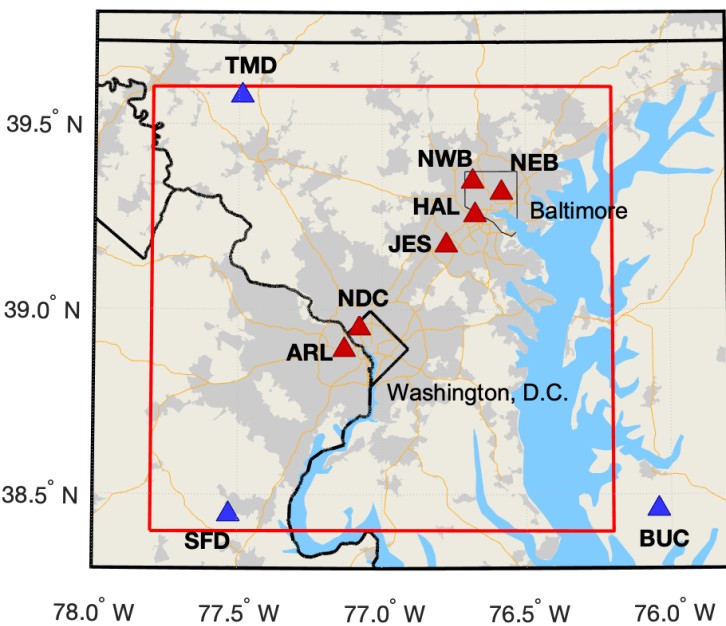

**Figure 1. Domain of interest for our study (red square), surrounding the metropolitan regions of Washington, D.C., and Baltimore, MD. Gray shading indicates U.S. Census-designated urban areas (www.census.gov). Red triangles indicate urban stations used in this study, and blue triangles indicate background stations. All map data layers obtained from either Natural Earth (naturalearthdata.com) or U.S. Government sources and are in the public domain.**

## 2.2 Transport model

Many of the methods we use for estimating $y_{BG}$ rely on a transport model simulation of the domain. We use the meteorological fields from the Weather Research and Forecast (WRF) model to drive the Stochastic Time-Inverted Lagrangian Model (STILT; (Lin et al., 2003)). Following Lopez-Coto et al. (2020a), WRF is configured with the RRTMG radiation scheme (Mlawer et al., 1997), Thompson microphysics scheme (Thompson et al., 2004; Thompson et al., 2008), Noah land surface model (Chen and Dudhia, 2001), the Kain-Fritsch cumulus scheme (for the 9 km domain only) (Kain, 2004), the 1.5- order closure scheme MYNN (Nakanishi and Niino, 2004, 2006) with the eddy mass-flux option (Olson et al., 2019) and the land-use classification from the 2011 National Land Cover Database (Homer et al., 2015). Three nested domains are used (9 km, 3 km and 1 km), with the innermost domain covering the urban area of interest, with 60 vertical levels with monotonically increasing thickness from the surface (34 levels below 3 km) and driven by initial and boundary conditions from the North American Regional Reanalysis (NARR) three-hourly data (Mesinger et al., 2006).





STILT generates influence functions, or footprints, that relate the enhancement measured at a given observation location to fluxes from an area at the surface. STILT also tracks mass-less particles backward in time, and here we use the particles from each observation to determine the location and time of exit from the domain (i.e., this is analogous to the location of entry of each air mass into the domain before eventually reaching the observation point). For this work, we emit 960 particles for each hourly mean observation from both urban and background towers. Particles are released over the entire hour to simulate hourly mean, and tracked back in time for five days. Footprints are calculated for two nested domains: an inner domain with a footprint gridded at 0.01 degrees (shown in Figure 1) and an outer domain with a footprint gridded at 0.1 degrees (Figure 2); the exit points of the particles are determined for both domains. The choice of the two domains was made specifically for our region, in order to capture large emissions sources and other urban areas outside Washington DC and Baltimore. The simulation time for STILT of five days was chosen so that most of the particles (over 90%) had exited the larger of the two domains by that time. The analysis covers the one-year time period from November 2016 through October 2017.

### 2.3 Sampling global model at the urban domain boundary

In the next few sub-sections we describe several methods for estimating the background that we investigate in this work (Table 1), beginning with methods relying on global model output.

**Table 1. Summary of background methods compared and evaluated for $CO_2$ and $CH_4$. References for all models can be found in the pertinent Methods section. *Models using Vulcan 3.0 as the flux in the near-field background use FFDAS 2015 and models using EPA use EDGAR v4.2 for the small region in Canada within our outer domain (Figure 2). **The right-most column indicates whether this background was evaluated in the synthetic data study (S; Sect. 3) or against actual (real) observations (R; Sect. 4).**

| Abbreviation | Type of background | $y_{BGfar}$ | $y_{BGnear}$ | $y_{BG}$ | Evaluation method** (S=synthetic, R=real) |
|---|---|---|---|---|---|
| **$CO_2$** | | | | | |
| **Global-CTE** | Global model sampling at inner domain boundary | | | CarbonTracker Europe | R |
| **Global-CT** | Global model sampling at inner domain boundary | | | CarbonTracker v2019 | R |
| **CT+V3+VPRM** | Nested background | CarbonTracker v2019 (1°x1°) | Vulcan 3.0* + VPRM (0.1°x0.1°) | $y_{BGfar} + y_{BGnear}$ | R |
| **CT+V3+CASA** | Nested background | CarbonTracker v2019 (1°x1°) | Vulcan 3.0* + CASA (0.1°x0.1°) | $y_{BGfar} + y_{BGnear}$ | R |
| **Upwind Lagged** | Observations from upwind site | | | Sampled at the time of air mass exit | S |





| Upwind Aft | Observations from upwind site | | | Mean afternoon average from same day | S, R |
|---|---|---|---|---|---|
| Upwind Column | Observations from upwind site | | | Sampled from a vertical column (profile) | S, R |
| **CH$_4$** | | | | | |
| Global-CAMS | Global model sampling at inner domain boundary | | | CAMS CH$_4$ v17r1s | R |
| CAMS+EPA | Nested background | CAMS CH$_4$ v17r1s (2°x3°) | EPA 2012* (0.1°x0.1°) | $y_{BGfar} + y_{BGnear}$ | R |
| CAMS+EDGAR | Nested background | CAMS CH$_4$ v17r1s (2°x3°) | EDGAR v5.0 2015 (0.1°x0.1°) | $y_{BGfar} + y_{BGnear}$ | R |
| Upwind Lagged | Observations from upwind site | | | Sampled at the time of air mass entrance | R |
| Upwind Aft | Observations from upwind site | | | Mean afternoon average from same day | R |
| Upwind Column | Observations from upwind site | | | Sampled from a vertical column | R |


In the global model method, a 4D field of the GHG mole fractions from an existing global model is sampled by each STILT particle as it exits (enters) the urban domain (Figure 1) at a given latitude, longitude, altitude, and time. Here, for CO$_2$ we use publicly available mole fraction output from two different global CO$_2$ inversion models: CarbonTracker (CT) version 2019 (http://carbontracker.noaa.gov (Peters et al., 2007; Jacobson et al., 2020)) and Carbon Tracker Europe (CTE (obtained by

request); (Peters et al., 2010)). These two are referred to as Global-CT and Global-CTE (Table 1). Both global models provide vertically-resolved, 3-hourly, 1-degree resolution CO$_2$ mole fraction fields. For CH$_4$, we use the Copernicus Atmosphere Monitoring Service (CAMS) global inversion 4D fields at 4-hourly, 2x3 degree resolution (v17rs1, available at https://apps.ecmwf.int/datasets/data/cams-ghg-inversions/ (Segers and Houweling, 2018)) (Global-CAMS). The advantage of sampling a global model as a background is that the mole fraction field varies in space and time, and this field is generated

from fluxes optimized using atmospheric observations. A disadvantage, however, is the global models' resolution is quite coarse relative to our small (~140 km across) domain and may not provide sufficient spatial resolution for the background (i.e., the entire domain is only slightly larger than one CarbonTracker grid cell).

## 2.4 Using a nested domain to define a two-component background

A second method of estimating a background is to use a nested domain and separate the background $y_{BG}$ from Eq. (1) into two

components (Eq. (2)).





$$y_{BG} = y_{BGfar} + y_{BGnear} \qquad (2)$$

The first component, $y_{BGfar}$, is obtained by sampling a global model as described above, but at a boundary far from the domain of interest (magenta boundary in Figure 2). The second component, $y_{BGnear}$, is determined from convolutions of STILT footprints with a flux field in the outer domain. The fluxes within the inner domain of interest are set to zero, so that $y_{BGnear}$ does not include any enhancements from the inner domain. It only represents enhancements from fluxes between the outer domain and the inner domain (Figure 2 shows examples of these fluxes).

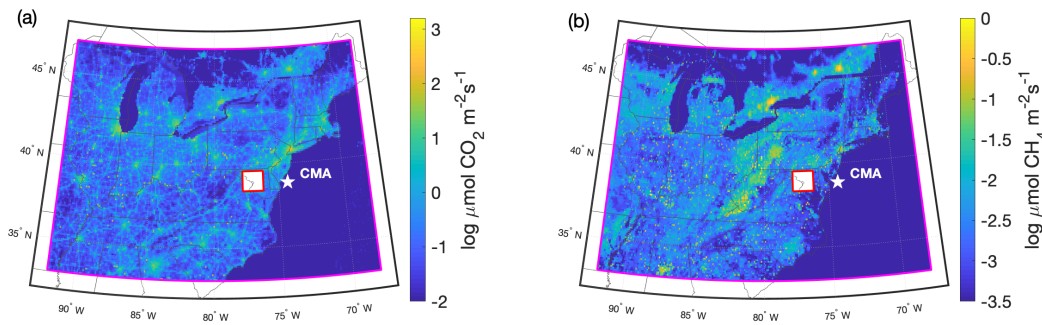

**Figure 2. Maps of nested domains used to calculate the two-component background. The outer domain (magenta) is used to determine the near-field background ($y_{BGnear}$) using footprints from STILT and existing flux inventories. (a) January 2015 mean of fossil-fuel $CO_2$ from Vulcan 3.0 with FFDAS in the Canada portion of the domain are shown in log scale. (b) January 2015 mean of the EPA $CH_4$ inventory with EDGAR in the Canada portion of the domain are shown in log scale. A global model is sampled at the edge of the outer domain for the far-field background ($y_{BGfar}$). The red square over Washington DC, and Baltimore, MD corresponds to the domain shown in Figure 1. The white star indicates the location of the NOAA aircraft site CMA. All map data layers obtained from U.S. Government sources and are in the public domain.**

One disadvantage to this two-component background is that, in our case, the fluxes used in the outer domain are not optimized using atmospheric observations; we rely on existing inventories or biospheric models. In addition, the existing anthropogenic inventories were developed for a different year than the study (for both $CO_2$ and $CH_4$), introducing additional uncertainty. However, the spatial resolution of the fluxes and meteorological model is better than for the global models (9 km for WRF and 0.1 degrees for the fluxes vs. 1 degree or more for the global models) and thus may better capture variability in background concentrations. We also can use different flux fields to estimate a range of background options using this method. For $CO_2$ we have used Vulcan 3.0 (Gurney et al., 2020b; Gurney et al., 2020a) for anthropogenic fluxes in the U.S. and the Fossil Fuel Data Assimilation System (FFDAS) (Asefi-Najafabady et al., 2014) in Canada. Both products are for the year 2015, and are adjusted to match the day of the week in the study year (2016/2017) (Figure 2a). We also use output from two biosphere models: a custom Vegetation Photosynthesis and Respiration Model (VPRM) (Gourdji, 2020), and an ensemble mean of the Carnegie-Ames-Stanford Approach (CASA) model run (Zhou et al., 2020) for biosphere fluxes (both for the time period of



our study). We refer to these two combinations as CT+V3+VPRM and CT+V3+CASA (Table 1). For $CH_4$, we have used the EPA 2012 gridded inventory (Maasakkers et al., 2016) (Figure 2b) and EDGAR v5.0 2015 (https://edgar.jrc.ec.europa.eu; (Crippa et al., 2019)) (referred to as CAMS+EPA and CAMS+EDGAR, respectively). We do not expect large biases in the anthropogenic $CO_2$ inventory fluxes at this regional scale, but the $CH_4$ and biosphere $CO_2$ fluxes are less well-known and may

introduce error. Specifically, this method is problematic for $CH_4$, where existing inventories have been shown to disagree significantly with measurements in the region upwind of our domain (Barkley et al., 2019), possibly due to the fact that the inventories are for different years than our study. One future goal of our project is to use inverse modeling to optimize fluxes in the outer domain to improve the accuracy of the background for the inner domain.

## 2.5 Using observations upwind of the urban domain, three different ways

Observations upwind of the domain of interest have been the most commonly used choice for background for urban studies (Lauvaux et al., 2016; Sargent et al., 2018; Nickless et al., 2018). The advantage to using observations over model-based estimates is clear: there is no need to depend on a global model or assume upwind fluxes are known. A background station also captures the variability in time that is expected of the background, but will not capture the variability in space, a consideration in this area with large spatial variability in upwind fluxes. In this study, we determine the upwind station as the

location that minimizes the difference between the mean particle exit angle and the angle to the background site. We choose between the stations in Thurmont, MD (TMD), Stafford, VA (SFD), and Bucktown, MD (BUC). In this section, we describe three ways to use measurements from an upwind measurement station, and then evaluate them for $CO_2$ in Sect. 3 using a synthetic data study. We choose the best method among these to evaluate along with model-based methods for both $CO_2$ and $CH_4$ in the real data study (Sect. 4).

### 2.5.1 Upwind Lagged method

We investigate using measurements from an upwind station in a truly Lagrangian fashion, i.e., to sample the upwind observations at the time an air parcel enters the domain of interest. This is typically not an effective method because at earlier times of the day, the mixing depth is often shallower than it is later in the day, and this method does not account for dilution of concentrations due to a growing planetary boundary layer (PBL). The background will be biased high and thus, the

enhancement determined at the urban tower would be negatively biased. In a synthetic data investigation of how to site background stations, Mueller et al. (2018) showed that although the upwind measurements sampled in this manner correlated well with the true background at the urban sites, they were biased high.

### 2.5.2 Upwind Afternoon method

A common method for overcoming the problem of diurnally varying boundary layer depth is to approximate the dilution in

concentration by using upwind observations at the same time as the observations at the urban site, when the PBL is similar





between the two (e.g., Lauvaux et al. (2016)). In our case, because we restrict our analysis to afternoon hours at the urban sites, this translates to sampling the upwind tower in the afternoon as well. This method must be considered carefully, and its effectiveness depends on the specific geography and location of the urban and rural measurement stations as well as the size of the domain. For example, on a summer day, a rural upwind tower at mid-day could be influenced by strong local

photosynthetic uptake causing a bias relative to the air measured at the urban tower at the same time; even if the same air mass passed the upwind tower it did so earlier in the day when there was less uptake. In a smaller domain, transit times to the boundaries are shorter in general and this effect may not cause much error. Otherwise, to alleviate the effect of these near-field fluxes when using a background observation at the same time as the urban observation, modeled enhancements (estimated using inventories inside the domain) from sources within the domain could be subtracted from the upwind concentration

(Lauvaux et al., 2016). However, if near-field fluxes outside the model domain influence the upwind towers (as is the case in our domain, because our background towers are either very close to the edge or outside the domain entirely), this correction may not entirely eliminate the problem.

### 2.5.3 Upwind Column method

This method accounts for dilution by free tropospheric air being entrained into the growing PBL by sampling the upwind

location using an ensemble of particle trajectories from STILT, as was done to sample the global model (Sect. 2.3). This method has been used previously in regional studies to sample an upwind curtain that was constructed using smoothed long-term measurements (Jeong et al., 2016; Karion et al., 2016). In those studies, the STILT particles were used to sample a mole fraction field (curtain) at the edge of the domain with latitudinal, vertical, and temporal variability. Unfortunately, in our case, we do not have enough upwind measurements to construct a full boundary curtain. Instead, we combine the idea of sampling

a background curtain using the particles' exit locations and times with the idea of sampling an upwind measurement station, similarly to Sargent et al. (2018). We construct vertical profile columns of $CO_2$ and $CH_4$ that do not vary laterally but allow the particles to sample a realistic vertical mole fraction gradient, and average the mole fractions in the column across the particles to calculate at the background value ($y_{BG}$). Below we describe the method for constructing vertical profile $CO_2$ and $CH_4$ columns at upwind sites for use with this method.

For every urban observation that we model using STILT, we construct a vertical profile, or column, background to sample with the particle trajectories. First, each particle is tracked back to its exit location from the domain, and the nearest background station is determined by comparing the exit angle and the angle between the background site and the urban station. If the nearest station does not have observations for the time that the particle exited, the next nearest is used. Until May 2017, only one background site was operational, BUC, meaning that backgrounds constructed using any of the upwind-observation-based

methods always use BUC until May 2017, when TMD was established. SFD was established in July 2017, so after that period all three stations were options. Note that in the synthetic data study, we use all three sites for the entire year as the ideal case



scenario, and then investigate the effect of using only one site without filtering for particular wind directions, as other studies have done.

Once the background station is identified, the modeled boundary layer height associated with the exiting particle's exit location
and time is used to construct a vertical profile $y(z)$ as shown in Eq. (3) and (4) and described below, where $y$ is the mole fraction in the column and $z$ is the altitude above ground level (AGL). We define two cases: one for afternoon hours (Eq. (3a) and (3b)) and one for non-afternoon hours (Eq. (3c) through (3e)); note that the time of day referred to here is the local time at which the particles exit the domain, not the time of the urban observation.

Afternoon hours:

$$y(z) = y_{obs}, \ z \leq PBL \tag{3a}$$

$$y(z) = y_{FT}, \ z > PBL \tag{3b}$$

Non-afternoon hours:

$$y(z) = y_{obs}, \ z \leq PBL \tag{3c}$$

$$y(z) = A + Be^{(-z/800m)}, \ PBL < z \leq 2000 \text{ m} \tag{3d}$$

$$y(z) = y_{FT}, \ z > PBL \tag{3e}$$

where the parameters A and B are constants calculated by imposing two boundary conditions on Eq. (3d):

$$y(z = PBL) = y_{obs} \tag{4a}$$

$$y(z = 2000) = y_{prevAFT} \tag{4b}$$

If the particle exited during afternoon hours (defined as five hours after sunrise and before sundown), then the profile represents
a two-layer troposphere consisting of the background site observation ($y_{obs}$) from the ground to the top of the PBL, and the free troposphere value, $y_{FT}$, (discussed below) above the PBL (Eq. (3a) and (3b)). If the particle exits during non-afternoon hours, the profile is constructed in three layers. The lowest layer, below the PBL, consists of the observation at the background tower at the exit time, $y_{obs}$ (Eq. (3c)). From the PBL to 2000 m AGL, the profile is assumed to be a residual layer and is modeled as an exponential decay function beginning with the tower observation ($y_{obs}$) at the PBL top to the concentration measured at that
same site the previous day (mid-day afternoon average), $y_{prevAFT}$ (Eq. (3d)). Above 2000 m AGL, the profile is based on the





free-tropospheric value $y_{FT}$ (Eq. (3e)). The height of the residual layer (2000 m) and the length scale of the exponential function (800 m) were determined using the synthetic experiment described in Sect. 3, by testing several values and choosing the best-performing combination (not shown). The choices for both of these values introduce error in the column background; for example, the height of the residual layer would change from day to day and here it is assumed constant.

The free-tropospheric mole fractions for all profiles ($y_{FT}$) are derived from binned and smoothed $CO_2$ and $CH_4$ observations from the National Oceanic and Atmospheric Administration's regular aircraft sampling at site CMA (Sweeney et al., 2015), available from the $CO_2$ GLOBALVIEWplus v5.0 ObsPack (Cooperative Global Atmospheric Data Integration Project, 2019). These observations are made on bi-weekly flights collecting whole-air samples in flasks at nine altitudes between 300 m and 8000 m above sea level, offshore and almost directly east of our domain (Figure 2). We assume that the CMA observations

above 2000 m are not influenced by fluxes in our inner domain and are representative of typical seasonally varying concentrations in the free troposphere above our domain. We bin the data into nine altitude bins between 0 m and 9000 m designed to evenly distribute observations between bins, and use the ccgcrv software from NOAA/ESRL (Thoning et al., 1989), available and documented at https://www.esrl.noaa.gov/gmd/ccgg/mbl/crvfit/crvfit.html, to smooth the timeseries within each altitude bin with 4 annual harmonics and 3 polynomial terms. Example profiles over BUC are shown in Figure 3.

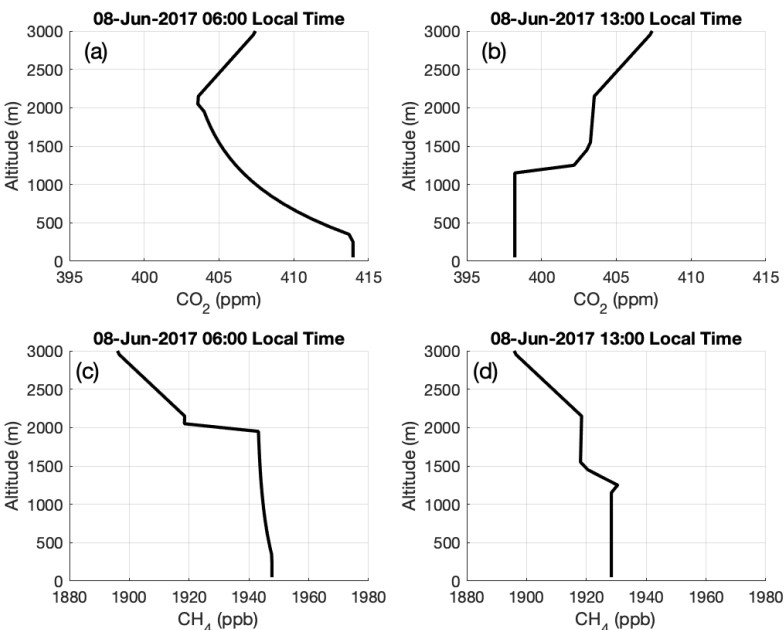


**Figure 3. Examples of $CO_2$ (a, b) and $CH_4$ (c, d) vertical column profiles above BUC for morning (a, c) and afternoon (b, d) on a summer day with winds from the east (i.e., when the site is upwind of the urban domain). Profiles are constructed as described in the text.**



## 3 Synthetic experiment evaluation of upwind observation-based $CO_2$ backgrounds

To evaluate the three upwind-observation-based $CO_2$ background conditions described in Sect. 2.5, a synthetic data experiment was devised similar to that described in Mueller et al. (2018). $CO_2$ was chosen rather than $CH_4$ because we believe we have a relatively realistic flux field to use for $CO_2$, whereas for $CH_4$, we find large differences between model estimates and observations. In particular, BUC is in an area with a large influence of local wetlands (Karion et al., 2020), so that the synthetic experiment would not yield necessarily realistic results without an accurate wetland model. The same-day afternoon sampling

of $CO_2$ is also more likely to be a problem due to strong biospheric fluxes in summer influencing observed concentrations at the background station; whether the column method alleviates this issue was a key question to answer with the synthetic experiment.

A set of synthetic $CO_2$ observations $y$ was constructed using the WRF-STILT footprints from our model for November 5, 2016 through October 30, 2017, for six urban sites (NWB, NEB, HAL, JES, NDC, and ARL) and all three background sites (BUC,

SFD, TMD) for the entire time period; see Figure 1 for locations). Note that in order to evaluate the effectiveness of the method, we simulated all three upwind sites for the entire year even though in reality two of them were established later in the year (May 2017 and July 2017 for TMD and SFD, respectively). The nested domain setup was used to construct observations for each afternoon hour at the urban sites (afternoon defined as the period between five hours after sunrise until sundown):

$$y = y_{BGnear} + y_{BGfar} + y_{enh}.$$  (5)

For the synthetic observations, $y_{BGfar}$ is derived by sampling CarbonTracker 2019 at the edge of the outer domain (Figure 2). $y_{BGnear}$ is derived from convolving WRF-STILT footprints in the outer domain with 2015 Vulcan 3.0 (Gurney et al., 2020b) (with FFDAS for the small Canadian portion of the domain) anthropogenic fluxes and VPRM (Gourdji, 2020), with zero fluxes in the inner domain. In other words, we construct the "true" background as CT+V3+VPRM as defined in Table 1. Although the anthropogenic flux data products are derived for the year 2015, they represent a plausible representation of sources in our

domain for this synthetic experiment. The enhancement from fluxes in the inner domain, $y_{enh}$, is the convolution of the footprints in the inner domain with Vulcan 3.0 and VPRM. Thus the true background, $y_{BG} = y_{BGnear} + y_{BGfar}$, is known for each synthetic observation $y$.

We also construct observations $y$ for all 24 hours at the background sites (BUC, TMD, SFD) in exactly the same manner, and use them to construct the synthetic Upwind Column background described in Sect. 2.5.3. For the synthetic column, free-

troposphere values are sampled from CarbonTracker v2019 (Jacobson et al., 2020) at the CMA location. Thus, the experiment assumes perfectly-known transport and perfectly consistent fluxes, and allows for the determination of how well a column background sampled above an upwind site represents the true background observed by the urban towers on any given afternoon hour. We also determine a background based on sampling the synthetic observations at the upwind site at the same time as the





urban site (i.e., Upwind Afternoon observations, as described in Sect. 2.5.2, with modeled in-domain enhancements removed)

and sampling the upwind site at a lagged time based on particle exit (i.e., Upwind Lagged observations, Sect. 2.5.1) to quantify

the biases in these three methods.

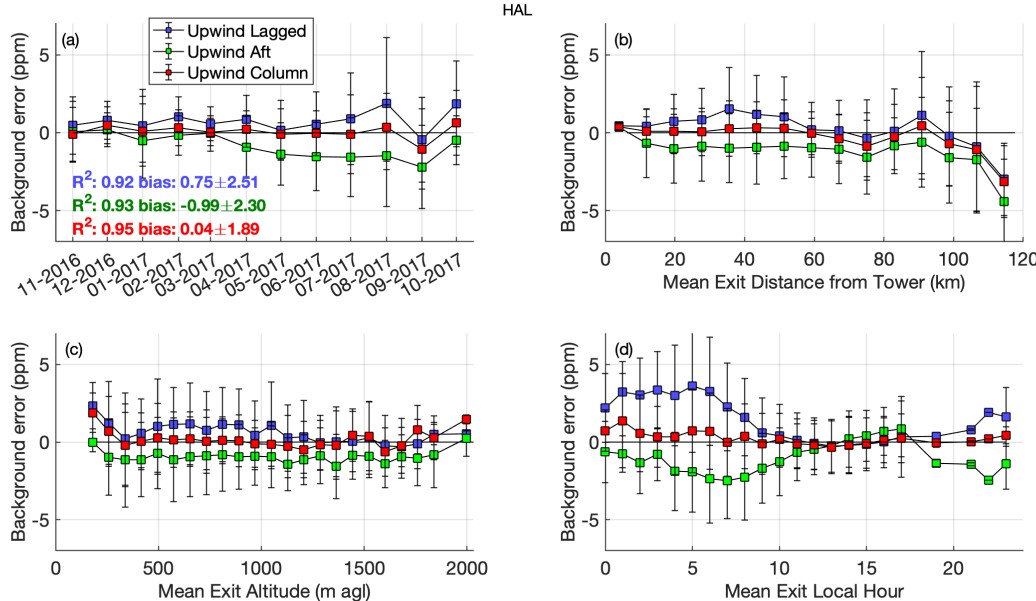

**Figure 4. Results of the synthetic data study. Error in background (constructed background – true background) at HAL (afternoon hours only) for each method of using upwind site observations described in the text. Results for the other urban towers are similar.**
**Red points indicate the column-based background (Upwind Col); green from sampling upwind sites at the same time as urban sites (Upwind Aft); blue from sampling upwind sites at the time of the particle exit (air mass entrance) earlier in the day (Upwind Lagged). Text in panel (a) indicates the coefficient of determination ($R^2$) and mean bias ± one standard deviation of the error over the entire year for each method in the corresponding colors. Panel (a) monthly mean, (b) binned by mean particle exit distance from the closest upwind tower, (c) binned by mean exit altitude, (d) binned by mean exit time of day. Error bars are standard deviations.**

We evaluate the error (defined as the true background subtracted from the background constructed using upwind observations)

by looking at the mean as a function of different factors: month of the year (Figure 4a), mean distance from background site

(Figure 4b), mean trajectory exit altitude (Figure 4c), and mean trajectory exit time of day (Figure 4d). The overall annual

statistics (mean bias, standard deviation, and $R^2$) (Figure 4a) indicate that the column-based background (red) is the best

performer. The results also indicate that sampling the upwind site at the time the air mass entered the domain (Upwind Lagged)

yields a high bias in the background (as described in 2.5.2) due to PBL dynamics (blue). Using the upwind observations from

the mid-afternoon (Upwind Aft) causes a summertime negative bias due to biospheric uptake (negative fluxes) near the upwind

tower (green). Figure 4d indicates that the largest errors in the non-column backgrounds occur when the air mass enters the

domain early in the morning, as is typical when using afternoon observations in this domain.





These results support using the upwind site observations at the same time as the downwind observations (Upwind Aft) if diurnally varying fluxes near the upwind tower are not a concern (for example, for fossil-fuel $CO_2$ only or wintertime only), or for instances where the domain is small enough that the transit time is short between the two stations. Otherwise, strong biosphere fluxes near the background sites that are unaccounted for can cause an overall summertime bias in the background at monthly scales. This conclusion may not be extensible to other network configurations (for example, depending on the location of background sites in relation to strong biological fluxes), but shows that for the network design here, sampling the background site at the same time as the urban site gives a biased background in summer. Figure 4c indicates that the biases in non-column methods occur when particles exit at higher altitudes, likely because these methods do not account for mixing of air from the free troposphere into the urban domain. However, they also show that the column-based background, as constructed here, does well at eliminating these biases. Figure 4 shows the results for one site (HAL) only, but the results do not vary much between sites (annual biases range from -0.02 ppm to 0.16 ppm; RMSE ranges from 1.81 ppm to 1.91 ppm).

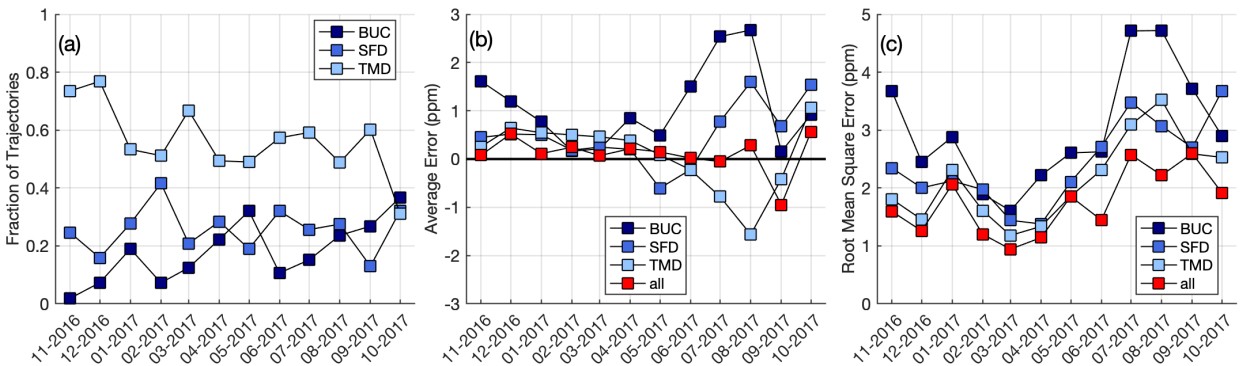

**Figure 5. Results from synthetic experiment. (a) Fraction of STILT particle trajectories exiting closest to each background tower by month (MM-YYYY). (b) Bias and (c) RMSE for column-based upwind backgrounds (relative to the true background) in the synthetic data experiment using only a single background site (shades of blue), compared with the ideal scenario of all three background towers having available observations (red). Average values over the six urban sites for each month are shown.**

As noted earlier, synthetic observations from all three background towers were used in this analysis, even though SFD and TMD were not established for some of the time period. Somewhat surprisingly, we do not see a large bias as a function of the distance between the exit trajectory and the upwind station below 100 km (Figure 4b), but a sharp increase in bias after that. Given that the distance between the trajectory exit and the designated upwind site should affect the error, we also investigated the bias and root-mean-square error (RMSE) for configurations in which only a single background site was available (Figure 5). Particle trajectory statistics from STILT indicate that most air masses enter the domain closest to TMD, the site in the northwest of the domain, with the fewest entering near BUC for most months of the year (Figure 5a), confirming that the predominant wind direction for this region is west or northwest. Both monthly biases and RMSE are generally larger when only using a single background site (Figure 5b and 5c); as one might expect, biases tend to be positive because the single site





may be downwind of the urban area for some time periods, depending on wind directions (e.g., BUC would be downwind
when winds are from the west, so observations there would be likely to be enhanced relative to the true background). The
RMSE might be further reduced if additional background towers were available; for our domain, specifically, we plan to
establish an additional background site in the northeastern corner of the domain. This site should better represent the
background when winds are from that direction (14% of the time), given the likelihood of elevated concentrations entering the
domain from upwind urban areas (e.g., Wilmington, DE, or Philadelphia, PA), which are not captured by the current
background stations.

## 4 Evaluation of $CO_2$ and $CH_4$ background performance using urban tower observations

The synthetic study described above is valuable in determining how to best use the upwind site observations to construct an
unbiased background. From that analysis, we conclude that the Upwind Column method performs best among the upwind
observation methods. However, there are several sources of error that are not accounted for in that setup. Specifically, errors
in transport (for example in the modeled PBL depth) would cause errors in the Upwind Column background, as would errors
stemming from the sparse sampling at CMA (which was binned and smoothed), which affect the free tropospheric value used
in the Upwind Column, while in the synthetic study those were modeled using CarbonTracker fields that are fully simulated
in space and time. Here we evaluate the Upwind Column method against the model-based methods described in Sect. 2 and
Table 1 against real observations of $CO_2$ and $CH_4$ from the urban sites. Because it is commonly used in urban studies, we also
evaluate the Upwind Afternoon method for comparison, even though we found it to be biased for $CO_2$ in the summer in the
synthetic study.

In this real-data comparison, we only use observations during times when we expect minimal contribution to the urban
enhancements from the urban domain. The goal is to isolate errors that are most likely to be caused by background choice
rather than the flux model inside the domain. To do this, we choose afternoon hours for which the magnitudes of the STILT
influence functions (footprints) are in the 10$^{th}$ percentile of all afternoon hours over the entire year-long study period, resulting
in 50 to 300 compared observations per month, with generally greater numbers in the summer months due to the longer
afternoon time period.

We calculate backgrounds for each urban site observation meeting the footprint strength criteria using multiple methods
described in Sect. 2 and summarized in Table 1, all utilizing the same WRF-STILT transport. We chose these as a set of
reasonable backgrounds; we also evaluated additional combinations for the nested methods but there was no significant
difference from those shown (e.g., choosing a different product for anthropogenic emissions for $y_{BGnear}$ or different global
model for $y_{BGfar}$). All combinations use the same fluxes inside the inner domain to calculate $y_{enh}$: Vulcan 3.0 and VPRM for
$CO_2$ and EPA (Maasakkers et al., 2016) for $CH_4$. Modeled inner domain enhancements for these observations range from 0



ppm to 7 ppm of $CO_2$ (2 ppb to 16 ppb $CH_4$) in any given month, with all months except November and January at or below 2

ppm ($CO_2$) and 6 ppb ($CH_4$). Error in the assumed fluxes inside the domain would affect these modeled enhancements and

contribute to the errors calculated in this analysis.

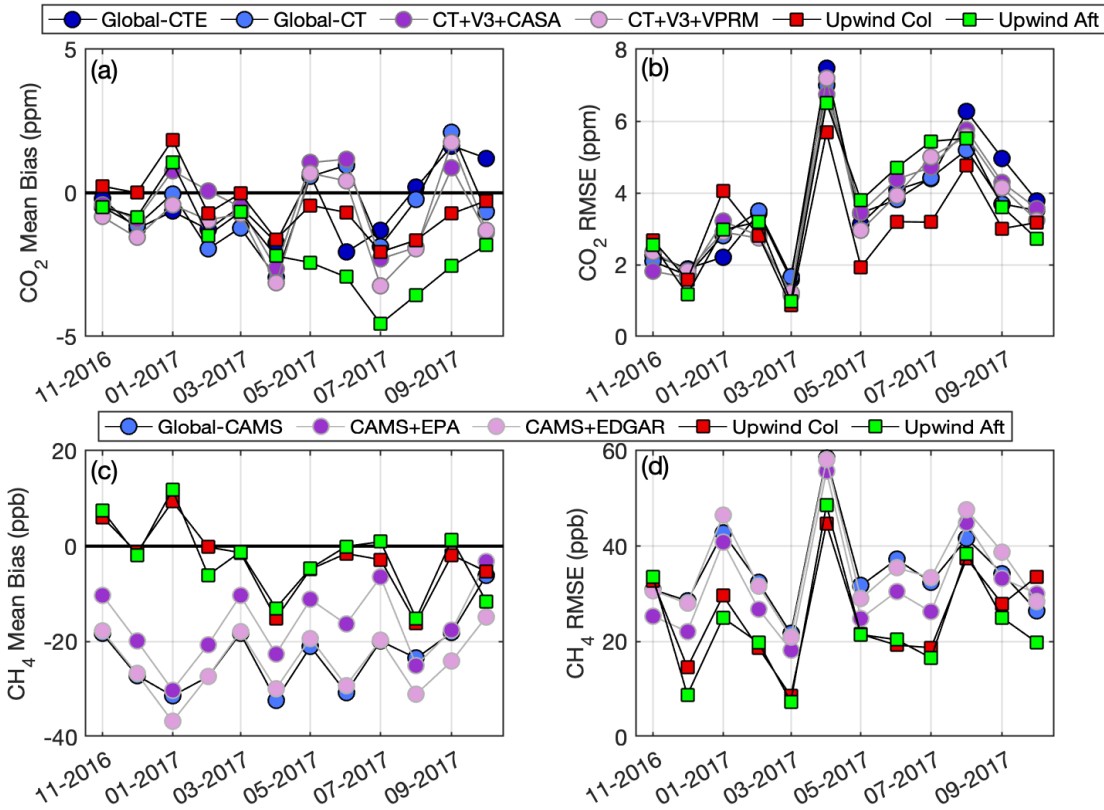

**Figure 6. Average monthly bias (Model – Observations) (a, c) and RMSE (b, d) for different backgrounds over all six urban sites during periods of low influence from within the domain of interest. (a, b) are for $CO_2$; (c, d) are for $CH_4$. All models use the same**
**fluxes inside the urban domain; only the background varies, as indicated in the legends above each set of panels, with abbreviations from Table 1.**

The model bias (modeled – observed) indicates that the Upwind Column background (red) performs well for $CO_2$, but is

negatively biased in the summer months (Figure 6a). Some positive bias in the upwind background is expected due to the lack

of upwind observations available from November through April from TMD or SFD. The synthetic data study had indicated

that using BUC alone leads to a high bias because it is not always upwind of the urban area (Figure 5), but in this evaluation,

only January has a positive bias using the Upwind Column background. There may be an offsetting negative bias; this and

some of the negative summertime bias may be caused by inaccuracy in the fluxes inside the domain (Vulcan 3.0 + VPRM)

rather than the backgrounds. This result suggests the possibility that the biosphere model is biased in the same direction (too





much summertime uptake or too little respiration), or that the error is not from the biosphere model. The anthropogenic
emissions in the domain could also be incorrect, affecting this result, and possibly offsetting a wintertime positive bias in the
background. The Upwind Aft background (green, Figure 6a) has an even larger negative bias in summer, a result consistent
with the synthetic data analysis. The root-mean-square-error (RMSE) indicates significant hourly variability (RMSE ranging
from 1 ppm to 8 ppm) in the background errors even when there is little bias (Figure 6b).

Methane results indicate that the four backgrounds relying on inventory or modeled emissions outside the domain have a
negative bias, while backgrounds based on upwind observations (both Upwind Col and Upwind Aft) are less biased throughout
the year (Figure 6c). RMSE analysis confirms that the upwind observation-based backgrounds perform better than the model-
based backgrounds for $CH_4$. Monthly variability of $CH_4$ RMSE follows similar patterns as for $CO_2$ (Figure 6b vs. Figure 6d),
for example in April 2017 both show large RMSE values, indicating that some of the error is likely from transport. Figure 6
shows statistics averaged over the six urban sites; monthly patterns in both bias and RMSE for each site are very similar to the
mean.



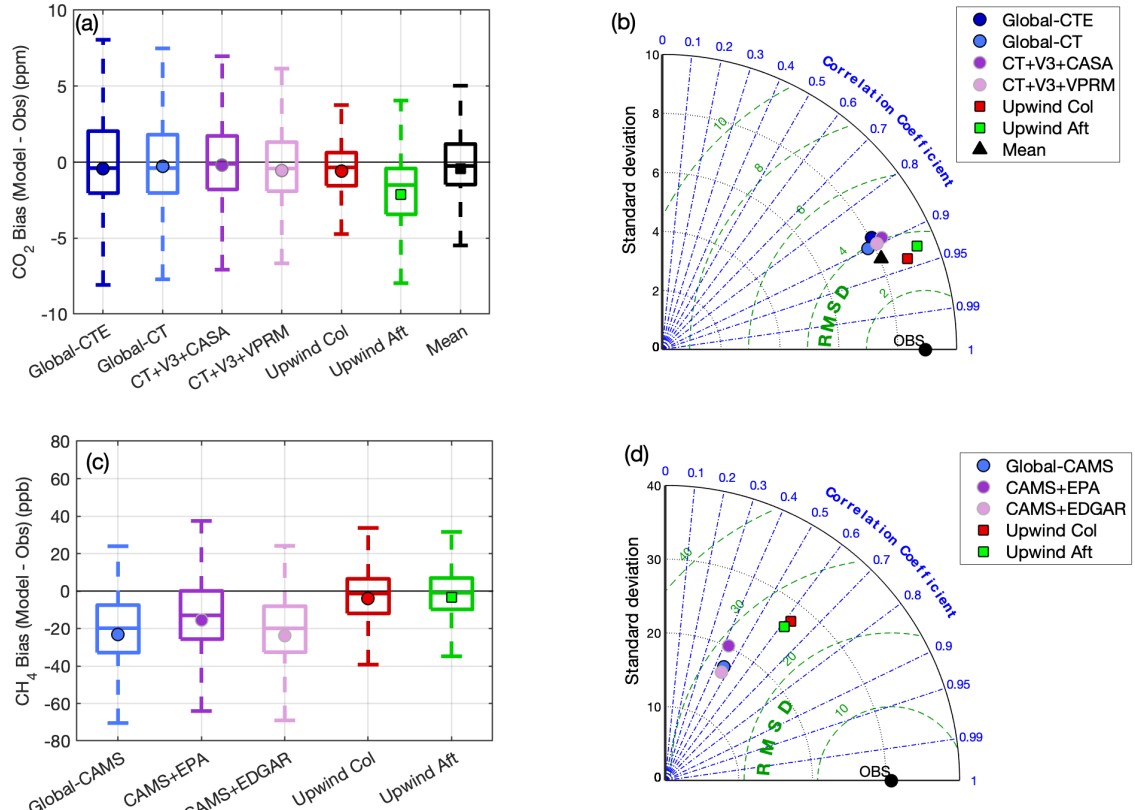

**Figure 7. Annual statistics for modeled vs. observed mole fractions over all six urban sites for $CO_2$ (a, b) and $CH_4$ (c, d) using identical fluxes in the inner domain with the different backgrounds from Table 1. For $CO_2$, we also include the mean of the first five (excluding Upwind Aft). (a, c) Bias (Model – Observations, afternoon hours); center line is the median value of the bias over all low-footprint hours of the year; symbol is the mean; box edges indicate $25^{th}$ and $75^{th}$ percentiles (inter-quartile range); whiskers show range excluding outliers; outliers not shown. (b, d) Taylor diagram (Taylor, 2001) illustrating performance replicating the standard deviation of the afternoon observations (black axes at constant radius), correlation (blue angular axes), and root mean square deviation (RMSD, green arcs).**

Analyzing the full year from all six urban sites together (all afternoon hours, Figure 7), for $CO_2$, the model-based backgrounds and the upwind column have close to zero net bias over the whole year, but the upwind column background performs best in terms of hourly scatter, as indicated by the smaller inter-quartile range in the box plot (7a). The $CO_2$ Taylor diagram (Taylor, 2001) in Figure 7b indicates that the correlation coefficient is quite high, close to 0.9 for all backgrounds, because they all successfully diagnose the seasonal cycle. The two backgrounds based on upwind observations perform best in terms of correlation coefficient, have lower root mean square deviations, and standard deviations closer to those of the observations (black circle on the x-axis), with the column background (red) performing best. We also evaluate the performance of a background that is the hourly mean of the first five backgrounds, i.e., excluding the Upwind Aft background which has a distinct low bias. This mean background performs fairly well, although not as well as the Upwind Column.



We evaluate five similarly constructed backgrounds for $CH_4$ (see Table 1 for specifics), and, just as in the monthly analysis (Figure 6c and 6d), find that the two backgrounds based on upwind observations perform best (Figure 7c and 7d). Unlike for $CO_2$, using the Upwind Afternoon observations (green) performs just as well as (even slightly better than, in terms of bias) the Upwind Column (red), with near-zero bias through the year. Both the bias box plots and Taylor diagram indicate that using an upwind observation for $CH_4$ is highly preferable to a background that relies on modeled emissions, because the models used

here (EPA gridded inventory (Maasakkers et al., 2016) and EDGAR v5.0 (Crippa et al., 2019) along with the global CAMS inversion (Segers and Houweling, 2018) are likely too low in their outer domain emissions. Correlation coefficients for $CH_4$ are significantly lower overall than for $CO_2$, even for the observation-based backgrounds, due to the lack of a strong seasonal cycle. Interestingly, the correlations are higher for the upwind backgrounds (coefficients close to 0.6) than for the model-based backgrounds (coefficients of 0.4 to 0.5); even though the model-based backgrounds might be assumed to better capture the

spatial variability of incoming air, that does not seem to be the case, because the poor quality of the emissions products used here negates this advantage.  The small negative bias of the Upwind Aft and Upwind Col backgrounds over the year (3 ppb and 4 ppb, respectively), are to be expected if emissions inside the domain are lower than the EPA 2012 inventory, which previous work suggests is the case (Lopez-Coto et al., 2020b).

**5 Discussion**

The large hourly variability of error in the background (as indicated in the inter-quartile differences shown in Figure 7) leads to the question of what the uncertainty is on enhancements from the urban network. This uncertainty is crucial to understanding the signal to noise ratio and is often required for any analysis, such as an atmospheric inversion.  Here for $CO_2$ we estimate the background uncertainty in two ways: one is to use the standard deviation of the first five backgrounds listed in Table 1 (i.e., omitting the Upwind Aft background, which we found to be biased in summer).  The second is to use the standard deviation

of the difference between modeled and observed $CO_2$ when using the best-performing background (in our case, Upwind Column as shown in Sect. 4) during times of low domain influence (i.e., the data shown in the boxplot in Figure 7a). These two quantities (shown as monthly means in Figure 8) have similar magnitudes in winter months, but the uncertainty estimated using the modeled to observation difference during low footprints (red) is up to twice as large in the summer. As this second method also includes uncertainty from the fluxes inside the domain, it may be an overestimate of the uncertainty. Note we

cannot estimate the uncertainty for $CH_4$ using the set of backgrounds in Table 1 as we do not believe the spread of these is realistic, since the four model-based backgrounds are clearly underperforming relative to the other two.



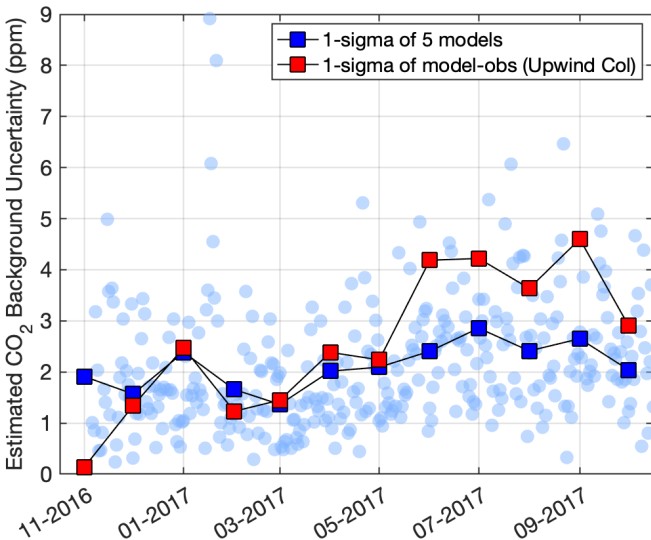

**Figure 8. Comparison of two methods for estimating uncertainty on CO$_2$ background at HAL. The blue squares indicate monthly means of the standard deviation of five different backgrounds (light blue circles are daily). Red points indicate the monthly means of the standard deviation of the difference between modeled and observed mole fractions using the Upwind Column background during low-footprint periods. The other sites show identical patterns and very similar values.**

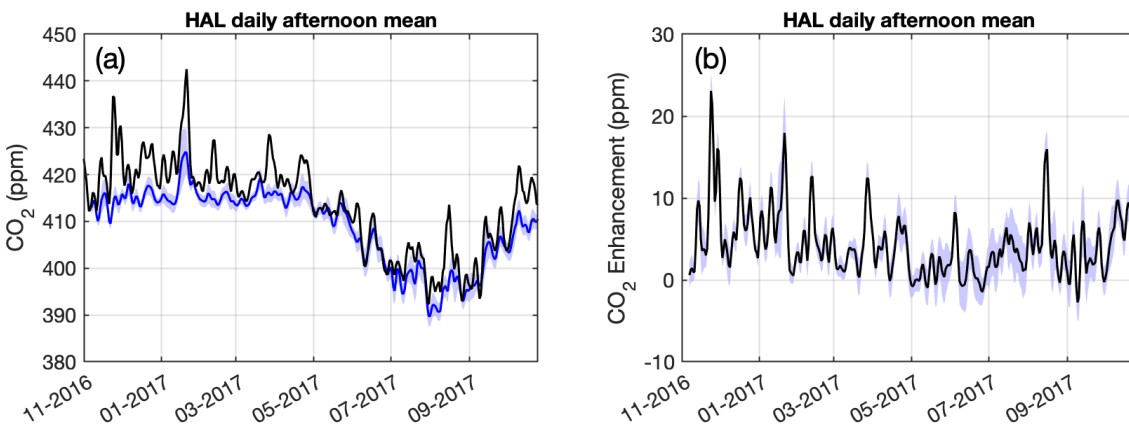

**Figure 9. (a) Observed CO$_2$ time series at HAL for one year (black) with the background (blue) as the mean of five backgrounds from Table 1. (b) Corresponding CO2 enhancement time series. In both panels, lines are the 7-day moving average of mid-afternoon daily means; blue shading indicates the standard deviation of the five backgrounds.**







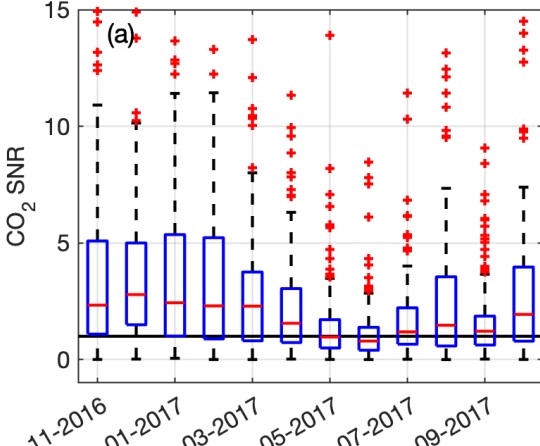
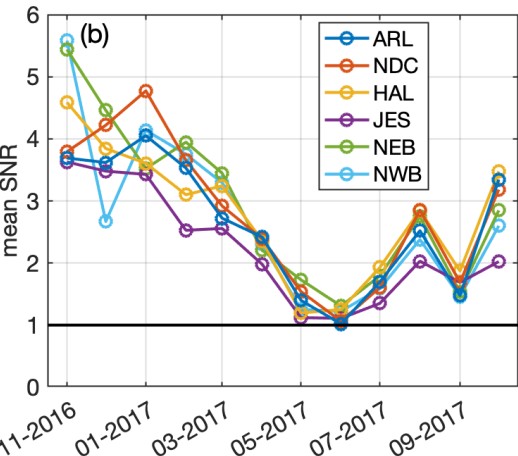

**Figure 10. (a) Monthly box plot of the hourly signal to noise ratio (SNR) at all sites (afternoon hours only), calculated as the $CO_2$ enhancement ($y_{enh}$) above the background divided by the standard deviation of the backgrounds (Figure 8) for each hour. Red lines are medians; box edges indicate 25th and 75th percentiles; whiskers indicate the extent excluding outliers, which are shown in red + marks. The y-axis has been truncated for readability so some outliers, up to 30, are not shown. (b) Average SNR by month for each site.**

We explore the impact of the background uncertainty on the ability of an urban measurement station to detect a signal in $CO_2$ enhancement. Figure 9a shows the background, chosen as the mean of the first five backgrounds we investigated in Table 1 (blue), along with observations (black), at HAL. We use the standard deviation of the five backgrounds, shown in blue circles and blue squares in Figure 8, as the uncertainty on each hourly background mole fraction (blue shading, Figure 9). Figure 9b shows the corresponding daily mean mid-afternoon enhancement (the background subtracted from the observed $CO_2$ mole fraction). The signal to noise ratio (SNR) is calculated as the ratio between the absolute value of the enhancement and the daily mean mid-afternoon standard deviation from the five backgrounds. Figure 10a shows the SNR boxplot for each month of the year for all sites together, while the mean SNR at each site is shown in Figure 10b. The analysis indicates that through the year there are periods, mostly in late fall and winter, when the observations show a clear enhancement above the uncertainty range of the background, and higher SNR. However, the median and mean SNR are low for much of the May to September time frame, because the enhancements over background are quite small during that time period, while background uncertainties are larger than in winter (Figure 8 and Figure 9b). Most of the loss of SNR is driven by small summer enhancements caused by lower anthropogenic emissions that are diluted by deeper planetary boundary layers and taken up by the significant vegetation in the domain. A similar result was found in Boston (Sargent et al., 2018), where summer enhancements were essentially zero in that similarly highly-vegetated metropolitan area. Estimating anthropogenic $CO_2$ fluxes in summer will thus be a challenge, requiring accurate modeling of both biospheric fluxes and meteorology to be able to overcome the uncertainty in the background conditions.



## 6 Conclusions

Previous work has shown that the background conditions in the Washington/Baltimore area have significant variability in both space and time (Mueller et al., 2018), as strong upwind sources of both $CO_2$ and $CH_4$ influence concentrations observed at urban tower sites. Here we compare a series of model-based backgrounds as well as backgrounds derived using upwind observations. Our evaluation against observations over one calendar year indicates that a background concentration derived from sampling observations from an upwind tower and accounting for vertical dispersion using a Lagrangian particle

dispersion model (such as STILT) performs well for both $CO_2$ and $CH_4$, with little bias over the year. But for $CO_2$, we find that our method may not be able to eliminate a summertime low bias entirely and some backgrounds based on sampling global or regional modeled concentrations at the edge of the domain perform almost as well. For $CH_4$, the conclusion is different: the less accurate regional and global modeled concentrations and the lack of strong diurnally varying biospheric fluxes near our background sites mean that using upwind observations (either using the vertical column or using same-time observations) as

a background is a better choice. Our analysis shows, however, that uncertainty on any individual hour or even month can be large, with summer mean monthly biases up to 2 ppm for $CO_2$ with significant scatter of 1 ppm to 3 ppm, and estimated random $CH_4$ uncertainties at 25 ppb (although this is likely an upper bound, as some of this scatter is from unknown fluxes inside the domain).

Our estimates urban enhancement uncertainty stemming from background uncertainty show that signal to noise ratios are small
in the Washington/Baltimore domain, drawing attention to the fact that background errors must be accounted for in any analysis of enhancements. This finding may not apply to a different urban region, for example a city with higher anthropogenic enhancements and smaller biospheric influence both within and outside its bounds. However, we believe the methods used here to evaluate different background products and assess uncertainty are extensible and can be applied in other urban and regional studies. One way to estimate background uncertainty is to use an ensemble of background methods and use their

spread as a proxy for the uncertainty in the chosen method. Our findings indicate that background choice is crucial. We recommend evaluation of background methods for a given urban domain, as the same background methodology may not be the best-suited for a different network design, region, or trace gas of interest.

## Data Availability

All observational data used in this work are archived at data.nist.gov and can be found at https://doi.org/10.18434/M32126
(Karion et al., 2019). CMA data are available from NOAA/GML at https://www.esrl.noaa.gov/gmd/ccgg/obspack/.



## Author contributions

Conceptualization by AK with input from KM, ILC, SMG and SG; investigation and analysis by AK; data by WC, EG, MS, and SP; model output from ILC (transport) and SMG (VPRM); writing by AK with review and editing by SMG, SG, ILC, KM, and JW. The authors declare no conflict of interest.

**Disclaimer**

References are made to certain commercially available products in this paper to adequately specify the experimental procedures involved. Such identification does not imply recommendation or endorsement by NIST, nor does it imply that these products are the best for the purpose specified.

## Acknowledgments

We acknowledge the assistance of the Earth Networks technical and engineering team, including Uran Veseshta, Clayton Fain, Bryan Biggs, Seth Baldelli, Joe Considine, and Charlie Draper for maintaining and operating the observational tower network. We thank Jooil Kim, Peter Salameh, and Kris Verhulst for assisting with data quality and software management. This work was funded by the NIST Greenhouse Gas Measurements Program. Support for Earth Networks provided by NIST grant numbers 70NANB15H344 and 70NANB14H322, and NIST commercial contract # 1333ND19PNB600853.

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
