# Peer review of "Background conditions for an urban greenhouse gas network in the Washington, D.C. and Baltimore metropolitan region"

_Atmospheric Chemistry and Physics, 2020_

## Referee Comment (RC1)

**Summary**

This paper analyzes different methodologies for calculating background values for $CO_2$ and $CH_4$ using a tower network in the Baltimore/DC area. The paper is very thorough, using various tower-based and model based methods combined with a Lagrangian approach to calculate the background. Results for the $CO_2$ portion behave as one might expect, with reasonable explanations provided throughout the paper and an ideal background methodology specific to that region determined by the end of the paper. Results for $CH_4$ also behave as expected, so long as expectations are built on a foundation of understanding that bottom-up $CH_4$ inventories are dreadful. While ultimately the results of this work are not surprising, they are **absolutely essential** in understanding the background variability for the DC/Baltimore region, and a necessary prerequisite to performing any sort of more complex emission quantification analysis using the tower network. To quote the final sentence of this paper "We recommend evaluation of background methods for a given urban domain, as the same background methodology may not be the best-suited for a different network design, region, or trace gas of interest." This paper does just that, and it does it well. **Publish with *very* minor revisions.**

**Line 49**: The abbreviation $CO_2$ is used here before it's defined on line 61.

**Line 166:** *"[CH₄] inventories have been show to disagree significantly with measurements in the region upwind of our domain (Barkley et al., 2019), possibly due to the fact that the inventories are for different years than our study"* While the increase of unconventional activity in the region since 2012 would create an underestimation of emissions, an updated EPA inventory would be nearly as wrong due to their flawed bottom-up inventory methods, as it thinks unconventional wells in that region have an average emission rate of 0.1% (hint: it's larger). Additionally, the EDGAR inventory you use is only about a year off of your analysis, so time isn't too much of an issue there. Personally I'd prefer a stronger statement on why the inventories are off. "*CH4 Inventories disagree with measurements, likely due to underestimations in oil and gas emissions inventories*". Skating around the fact feels like a disservice to the billions of studies screaming into the void that bottom-up oil and gas inventories are too low. We know the inventory is wrong.

**Section 2.5.2 Afternoon tower method:** May also be worth mentioning as a concern that on days with more complex wind patterns, the upwind tower may not even represent the same airmass as the downwind tower (i.e. frontal crossings or stagnant winds).

**Line 216:** *"First, each particle is tracked back to its exit location from the domain, and the nearest background station is determined by comparing the exit angle and the angle between the background site and the urban station. If the nearest station does not have observations for the time that the particle exited, the next nearest is used. Until May 2017, only one background site was operational, BUC, meaning that backgrounds constructed using any of the upwind-observation-based 220 methods always use BUC until May 2017, when TMD was established. SFD was established in July 2017, so after that period all three stations were options. Note that in the synthetic data study, we use all three sites for the entire year as the*

*ideal case, and then investigate the effect of using only one site without filtering for particular wind directions, as other studies have done."* I may just be confused here, but I would think that all of this is relevant to all your tower-reliant background methods, not just your upwind column one. If so, it feels out of place as the end of the previous paragraph (Line 214) makes it sound like this process and missing tower sites are exclusive to the upwind column method. I figured it out eventually, but it could probably be arranged better.

**Lines 239-end of section:** Not having gotten to the results yet, I just want to say I hope the column method is the worst because I don't want to have to replicate it on all my tower studies. But it's going to be the best, isn't it, or else I wouldn't be reading about it?

**Line 267:** "*whereas for CH4, we find large differences between model estimates and observations*". Wetlands would definitely be a problem for BUC, but was it still unsalvageable when winds have a westerly component and BUC would be irrelevant?

**Line 416**: "*Unlike for CO2, using the Upwind Afternoon observations (green) performs just as well as (even slightly better than, in terms of bias) the Upwind Column (red)*" Thank goodness

**Line 424**: *"even though the model-based backgrounds might be assumed to better capture the spatial variability of incoming air, that does not seem to be the case, because the poor quality of the emissions products used here negates this advantage".*

So one rather significant problem with your model background approach for CH4 is that neither EDGAR nor EPA inventories contain anything for wetlands. So that's dangerous, and perhaps part of the reason for the poor spatial correlation with the model-based backgrounds, as some wetland maps show wetland emissions in your large domain to play a substantial role in the concentration field, both with emissions on the east coast and emissions from Canada. I could make your life hell and say "redo this using one of the 250 WetCHART ensemble members included in the flux", but honestly I've never seen any of them actually produce anything resembling improvement to model vs obs comparisons. So in the end, it might be best to just mention that your EPA and EDGAR approaches are missing wetland emissions, which could in part explain why the upwind tower approach seems to do better, but there's little that can be done about it (and maybe even an argument that an upwind tower is necessary for CH4 since we can't adequately model a major source of CH4 spatially or temporally).

---

## Author Comment (AC1)

Response to Reviewer 1, Zachary Barkley.

Reviewer comments are in italics text; responses in plain text.

*Summary*
*This paper analyzes different methodologies for calculating background values for $CO_2$ and $CH_4$ using a tower network in the Baltimore/DC area. The paper is very thorough, using various tower-based and model based methods combined with a Lagrangian approach to calculate the background. Results for the CO 2 portion behave as one might expect, with reasonable explanations provided throughout the paper and an ideal background methodology specific to that region determined by the end of the paper. Results for CH 4 also behave as expected, so long as expectations are built on a foundation of understanding that bottom-up CH 4 inventories are dreadful. While ultimately the results of this work are not surprising, they are absolutely essential in understanding the background variability for the DC/Baltimore region, and a necessary prerequisite to performing any sort of more complex emission quantification analysis using the tower network. To quote the final sentence of this paper "We recommend evaluation of background methods for a given urban domain, as the same background methodology may not be the best-suited for a different network design, region, or trace gas of interest." This paper does just that, and it does it well. Publish with very minor revisions.*

We thank the reviewer for the thorough reading and comments that have improved and clarified the manuscript.

*Line 49 : The abbreviation CO 2 is used here before it's defined on line 61.*

Fixed.

*Line 166: "[CH 4 ] inventories have been show to disagree significantly with measurements in the region upwind of our domain (Barkley et al., 2019), possibly due to the fact that the inventories are for different years than our study" While the increase of unconventional activity in the region since 2012 would create an underestimation of emissions, an updated EPA inventory would be nearly as wrong due to their flawed bottom-up inventory methods, as it thinks unconventional wells in that region have an average emission rate of 0.1% (hint: it's larger). Additionally, the EDGAR inventory you use is only about a year off of your analysis, so time isn't too much of an issue there. Personally I'd prefer a stronger statement on why the inventories are off. " CH4 Inventories disagree with measurements, likely due to underestimations in oil and gas emissions inventories ". Skating around the fact feels like a disservice to the billions of studies screaming into the void that bottom-up oil and gas inventories are too low. We know the inventory is wrong.*

Thanks for this comment, this is a good point. We wanted to point out the difference in the year, but as the reviewer suggests, this is not necessarily the reason for the poor performance, and the cited study supports that this is an issue with the oil and gas sector. We made the change but also kept the sentence noting the difference in year, which we believe is still worth noting.

*Section 2.5.2 Afternoon tower method: May also be worth mentioning as a concern that on days with more complex wind patterns, the upwind tower may not even represent the same airmass as the downwind tower (i.e. frontal crossings or stagnant winds).*

Added.

*Line 216: "First, each particle is tracked back to its exit location from the domain, and the nearest background station is determined by comparing the exit angle and the angle between the background site and the urban station. If the nearest station does not have observations for the time that the particle exited, the next nearest is used. Until May 2017, only one background site was operational, BUC, meaning that backgrounds constructed using any of the upwind-observation-based 220 methods always use BUC until May 2017, when TMD was established. SFD was established in July 2017, so after that period all three stations were options. Note that in the synthetic data study, we use all three sites for the entire year as the ideal case, and then investigate the effect of using only one site without filtering for particular wind directions, as other studies have done." I may just be confused here, but I would think that all of this is relevant to all your tower-reliant background methods, not just your upwind column one. If so, it feels out of place as the end of the previous paragraph (Line 214) makes it sound like this process and missing tower sites are exclusive to the upwind column method. I figured it out eventually, but it could probably be arranged better.*

Thanks for this comment, these sentences have now been moved up into 2.5 where we describe the process for all three upwind methods.

*Lines 239-end of section: Not having gotten to the results yet, I just want to say I hope the column method is the worst because I don't want to have to replicate it on all my tower studies. But it's going to be the best, isn't it, or else I wouldn't be reading about it?*

Yes – but only for CO2! We found that the column method did not improve the background for CH4. More on this on a later comment.

*Line 267: " whereas for CH4, we find large differences between model estimates and observations ". Wetlands would definitely be a problem for BUC, but was it still unsalvageable when winds have a westerly component and BUC would be irrelevant?*

We would not say the OSSE was unsalvageable for CH4, it definitely still worked, we just did not feel we could trust the results since the entire OSSE requires that we think the setup is realistic, meaning we think the "true" fluxes we assumed are close to reality. That would be needed in order to trust that the results tell us something about the real situation here. This was not the case for CH4 in general, even if wetlands were not being considered. We could still do the OSSE, we just chose CO2 as a better case.

*Line 416 : " Unlike for CO2, using the Upwind Afternoon observations (green) performs just as well as (even slightly better than, in terms of bias) the Upwind Column (red)" Thank goodness*

Indeed.

*Line 424 : "even though the model-based backgrounds might be assumed to better capture the spatial variability of incoming air, that does not seem to be the case, because the poor quality of the emissions products used here negates this advantage".*

*So one rather significant problem with your model background approach for CH4 is that neither EDGAR nor EPA inventories contain anything for wetlands. So that's dangerous, and perhaps part of the reason for the poor spatial correlation with the model-based backgrounds, as some wetland maps show wetland emissions in your large domain to play a substantial role in the concentration field, both with emissions on the east coast and emissions from Canada. I could make your life hell and say "redo this using one of the 250 WetCHART ensemble members included in the flux", but honestly I've never seen any of them actually produce anything resembling improvement to model vs obs comparisons. So in the end, it might be best to just mention that your EPA and EDGAR approaches are missing wetland emissions, which could in part explain why the upwind tower approach seems to do better, but there's little that can be done about it (and maybe even an argument that an upwind tower is necessary for CH4 since we can't adequately model a major source of CH4 spatially or temporally).*

We agree with this assessment, and we agree that wetland emissions do affect our eastern boundary as well as possibly the eastern portion of our domain. We chose to exclude them in the analysis because of what the reviewer points out – we do not have a good wetland flux model. We do not think that there is a strong wetland signature at the urban or other two upwind towers, however, so we still think most of the poor performance of model-based backgrounds is an overall underestimate of emissions both to the west and inside the domain. However, this is a very good point and we have added it to the discussion here.

---

## Author Comment (AC2)

Response to Reviewer 2, Grant Allen.

Reviewer comments are in italics text; responses in plain text.

*Summary:*

*The paper is a very thorough examination of various model-and-measurement-based approaches to establishing upwind urban-background mole fractions of methane and carbon dioxide for use in Lagrangian (anthropogenic) flux calculations using urban measurement network measurements. It rightly highlights the significant challenges that such methods are typically (if not always) subject to, which include PBL mixing/dilution, biospheric attenuation, significant problems with inventories/priors, global model backgrounds, and model transport error. Some key conclusions of the paper are that an upwind column method appears to provide optimal backgrounds, but with caveats that summertime presents (expected) challenges concerning temporal PBL development and potential negative biases due to biospheric influences. These conclusions are not at all surprising, but they are very useful to others following this work and attempting to conduct urban GHG flux closure. The results are highly specific to the Washington/Baltimore area but the authors are very upfront about that and rightly suggest that conditions need to be assessed on a case-by-case basis. I would also assume that the uncertainties presented here are highly specific to the design of the measurement network/sampling (and this environment) and cannot be taken to be more broadly representative for other areas – and the paper does point this out.*

*The paper is generally very well written and well-presented. It represents incremental scientific advancement is that is very important to others attempting similar important work. I recommend publication after only small potential modifications and perhaps some thought to the specific comments below which may help increase the impact of the paper for others.*

We thank the reviewer for the thorough reading, analysis, and suggestions for improving the manuscript.

*Specific comments:*

*1/ I think the conclusions section could be clearer than it is. There is no central guidance in the conclusions section on what method(s) – I.e I think the upwind column method? – is/are optimal, and which are highly sub-optimal. Summarising some of what was concluded in the discussions section would be very useful, and arguably more important than discussing the RMSE and bias errors, which are highly specific to this singular environment and case study. Moreover, I wasn't entirely clear after reading the paper whether the UCF might be optimal in all conditions/seasons, or whether other methods may be better under specific conditions?*

We agree and we have now added some statements to clarify our recommendations and findings in the conclusion section, first paragraph. We specifically note that the upwind-afternoon background performs well for $CO_2$ in winter and all year for $CH_4$. The upwind column background performs better for $CO_2$ in summer and equally well for $CH_4$ and winter $CO_2$. We also point out in the 2nd conclusion paragraph that model-based backgrounds seemed to be

unbiased and perform well (although not as well as the upwind-observation-based) for $CO_2$, but not at all well for $CH_4$.

*2/ The paper could offer more guidance on what the authors consider might be an optimal network design in future, especially concerning how to place upwind measurement stations. Given that the central conclusion here is that a measurement-based background is optimal (I hope I've read that correctly?), can you go any further here to talk about whether model-based backgrounds should ever be trusted/useful, and/or whether towers with measurements at more/various heights might aid background, especially considering the biospheric problem where sinks are obviously land-based – for example, could a mix of surface sites and towers go some way to addressing the biospheric problem? It seems to me that upwind surface measurements may be more important than anything else here. Residual layers at higher altitudes are of course also important, but I would imagine that after ventilation from the day before, upwind surface measurements and free tropospheric knowledge may be more important than vertically-resolved measurements all through the background PBL?*

We thank the reviewer for this suggestion and agree that our study should provide some guidance on network design and additional useful measurements. We have now added an additional paragraph to the Conclusion:

"Our study allows us to give some guidance with regard to background for researchers establishing urban GHG tower networks. First, establishing stations upwind of the area of interest in a configuration that has been shown to capture incoming air from the predominant wind directions is crucial. For our network, a synthetic design study by Mueller et al. (2018) identified locations whose observations best correlated with "true" background. Second, the best-performing background for summertime $CO_2$ required integrating the upwind tower observations with knowledge of boundary layer height and observations in the free troposphere. We used existing free tropospheric observations from the NOAA/GML aircraft network, which provided measurements every two weeks at best. More frequent observations would have better captured synoptic-scale variability above the PBL and likely improved the Upwind Column background. Some capacity to conduct such airborne measurements should be considered in urban studies. Third, model-based backgrounds should still be considered, especially in cases where they can either be optimized in the urban inversion directly or informed by a nested inversion framework that allows upwind fluxes to be estimated rather than assumed. We did not extend our study to optimizing the modeled backgrounds using the tower observations, but it would be one way to adjust the modeled background and improve performance."

On the second point, we agree with the reviewer's thinking that additional observations aloft to help inform the column-based background would be very useful, even for our study where we relied on relatively sparse existing NOAA flask measurements. This information helps construct a better vertical column. We agree with the latter point the reviewer makes, in that we do not believe that additional vertically resolved measurements within the PBL will change the background much, because many air parcels enter the domain above the PBL and currently the FT is probably the least accurate part of the column.

*3/ Many of the biospheric problems discussed here exist simply because of the way the goal is defined, which is to deconvolve anthropogenic flux components. However… this problem could be turned around if the goal is about understanding "net" urban GHG emissions. Given the growing agenda on Net Zero carbon, urban biospheres are an integral part of the equation/solution. You could argue that understanding net emissions are an important result in and of itself. It might be interesting to discuss the pros and cons of this. The surrounding biosphere creates a problem for establishing representative upwind backgrounds (fully recognised and discussed), but the urban biosphere also creates a problem for deconvolving anthropogenic urban emissions – I don't think the latter problem is recognised or mentioned in the paper, but there are also good reasons why we might want to know what it is in a "net" sense.*

We do not agree with the specific premise of the first statement here, i.e., that the biospheric problems in our study exist simply because of the way the goal is defined. Rather, we agree with the statement later in this paragraph: the biospheric problems discussed here are those that confound the use of an upwind tower observation as a background in summer for $CO_2$.  This issue is caused by biospheric fluxes near the upwind towers, and is not related to urban biospheric fluxes inside the domain. Whether we are solving for a "net" enhancement in the domain or just the anthropogenic enhancements, this is still a problem, as it affects the calculation of the total enhancement, whether that is positive or negative, and independent of the goal of the study in terms of partitioning any net $CO_2$ enhancement between anthropogenic or biospheric components.

We fully agree that the urban biosphere does indeed create a problem for deconvolving anthropogenic emissions, and the reviewer is correct that we do not discuss this at all in this study. We do discuss the impact it has on the SNR (and perhaps this is the problem the first statement is referring to), because it causes small enhancements in summer, and we have added a sentence in the legend of Fig. 9 to make clear that the enhancements shown are "net", and not fossil-only. We also added a sentence in the discussion of SNR below figure 10 to point out that the SNR would be larger in summer if only FF enhancements were considered, for example if a biosphere model was used to determine fossil enhancements alone (of course, this would introduce error on that correction due to improperly modeled biological fluxes).

As the reviewer points out, we do not really focus on whether the goal is to diagnose emissions from a particular sector or process.  However, the reviewer makes a good point that this could be mentioned in the text so we have now added a sentence to point this fact out (last sentence of the discussion now, clarifying that while the background determination is challenging in summer, it is additional to the challenge of separating biospheric from anthropogenic fluxes within the domain).

*4/ In the conclusions section, a suggestion is posed on using "ensembles" of the different methods/models as a proxy for error/uncertainty. I would  disagree with this. This would not really be an ensemble, as each method/model is systematically entirely different. Ensembles usually represent variations (e.g. monte carlo simulations or parameter space) of a systematically-consistent approach (e.g. perturbing winds, prior uncertainty space etc in one model/method). You wouldn't really get a statistically-relevant ensemble by comparing apples*

*with pears this way, and it's a very different approach to e.g. comparing outputs of different climate models (which the IPCC would call an ensemble). I'm not even convinced it would give a max/min range of uncertainty that could be useful. I can see why it's attractive to comment on how uncertainty may be better defined but I'm far from convinced that the above would be fit-for-task. I would recommend removing this, or if not, then to discuss the above caveats or suggest alternative guidance on how to establish uncertainty. But this is just a suggestion.*

We pondered over how to define an uncertainty on the background for this study – after investigating the various backgrounds and evaluating their performance, we believe that a reader who might be interested in this topic would really just want to know the uncertainty on the background value. This uncertainty could then be incorporated into the model-data mismatch or observational error in an inversion for example.

We agree that the spread of several very different background representations does not represent the true uncertainty; this is unknown. But we do believe that the five realistic representations we have chosen can indeed be used as an ensemble whose spread may be used as a proxy for uncertainty, because they are all different realizations of the same physical system (the background value). We have already evaluated these five as best we can and believe they are unbiased and well-performing according to various metrics, and specifically we do not carry out this exercise for $CH_4$, where we know that four of the six backgrounds are not realistic. We also have shown that in winter, the magnitude of the model spread (mean over a month) in winter is similar to the error we found by comparing with observations (Fig. 8), giving us confidence that the spread of the five backgrounds is not unrealistically small or large. Thus, we have chosen to retain the comparison of model spread to the enhancement (the SNR in the Discussion), as we still believe this is a useful comparison.

However, in order to address the reviewer's concern and clarify to the reader the above issues and caveats, we have now added text at the beginning of the Discussion to add the caveat about this not being a true probabilistic ensemble, as follows:

"Unfortunately, the true uncertainty of the background is unknown. However, we can observe the differences between the various realistic and plausible representations of the true background that we have constructed for $CO_2$. We limit this set of plausible backgrounds to the first five backgrounds listed in Table 1 (i.e., omitting the Upwind Aft background, which we found to be biased in summer). Although this set of five background time series does not represent a formal probabilistic ensemble, the spread of these members can still inform us as to the confidence we have in any one of them or their mean."

Also, as the reviewer suggested, we have removed from the Conclusion the recommendation of using different backgrounds to define an "ensemble" as a proxy for uncertainty.

*5/ The uncertainties are presented in concentration space (for background). But nothing is given in terms of how this may manifest as flux error. I guess SNR is a proxy for this to some extent and I guess this paper's scope is on background evaluation, so I don't strongly suggest that flux error should be included, but if there is anything you can say about that, it may be helpful.*

We believe that translating background error into flux error is indeed beyond the scope of the work here. We did include the SNR analysis for the purpose of trying to understand the impact of background errors on the enhancement uncertainty which in turn impacts flux errors. We also focus much of the evaluation of the various backgrounds on bias because bias will have the largest impact on flux errors from atmospheric inversions. We have now added the following sentence at the end of the Conclusions:

"We specifically focus our evaluation metrics on bias, as biases will have the largest impact on posteriors from atmospheric flux inversions (as compared with random errors)."

*Technical corrections: None – thank you.*